# Design of a hesitant movement gesture for mobile robots

**Jakob Reinhardt** *, Klaus Bengler

Chair of Ergonomics, Department of Mechanical Engineering, Technical University of Munich, Munich, Bavaria, Germany

* jakob.reinhardt@tum.de

**Data Availability Statement:** All relevant data are within the manuscript and its Supporting information files.

**Funding:** The authors received no specific funding for this work.

## Abstract

In previous experiments, a back-off movement was introduced as a motion strategy of robots to facilitate the order of passage at bottlenecks in human-robot spatial interaction. In this article we take a closer look at the appropriate application of motion parameters that make the backward movement legible. Related works in distance perception, size-speed illusions, and viewpoint-based legibility considerations suggest a relationship between the size of the robot and the observer's perspective on the expected execution of this movement. We performed a participant experiment ($N = 50$) in a virtual reality environment where participants adjusted the minimum required back-off length and preferred back-off speed as a result of the robot size, and the viewpoint of the back-off movement. We target a model-based approach on how appropriate back-off design translates to different sized robots and observer's viewpoints. Thus, we allow the application of back-off in a variety of autonomous moving systems. The results show a significant correlation between the increasingly expected back-off lengths with increasing robot size, but only weak effects of the viewpoint on the requirements of this movement. An exploratory analysis suggests that execution time might be a promising parameter to consider for the design of legible motion.

## 1 Introduction

The use of robot motion to convey intentions to the surrounding environment was identified as a viable way to defuse spatial conflicts in human-robot interaction [1]. A movement is said to be legible if an observer can infer a robot's intention efficiently and confidently from observing its unobstructed and undistorted movement [1, 2]. For this purpose, a human-inspired back-off and hesitation movement has been implemented into an industrial robot's control strategy [3, 4]. This motion strategy was subsequently adopted for mobile robots to facilitate the order of passage at bottlenecks in human-robot spatial interaction [5]. In this earlier work, we highlighted the prerequisites regarding human abilities to perceive the back-off and assessed its impact on human movement efficiency. Two versions of a back-off movement were implemented. In a short back-off movement, the robot moved backwards for 1 s at a maximum speed of 0.2 m/s, reaching a back-off length of 0.19 m before stopping again. In the other condition, the robot drove backwards for 3 s and reached a back-off length of 0.54 m with the same

**Competing interests:** The authors have declared that no competing interests exist.

maximum speed. Reinhardt et al. (in press) [5] left open what combination of distance traveled, speed, execution time, and other properties such as the size of such a robot will result in a harmonious picture in the eyes of an observer. The aim is to meet the demand for a legible but also efficient movement strategy. Therefore, it must be sufficient to communicate the intention to yield priority and at the same time demand minimal path irregularities for a robot. To enable the use of back-off as a general motion language for robots, it would be necessary for developers to understand how the design requirements for a back-off motion can be transferred to robots with different characteristics. Accordingly, the main contributions of this article are:

1. A guide to setting the parameters, speed, and distance traveled for a legible back-off motion of a robot.

2. A model-based approach on how appropriate back-off design translates to different sized robots and observer viewpoints.

Motion design was emphasized early on in the animation discipline. Lasseter et al. (1987) [6] formulated fundamental principles for motion design in screen-based animation. Certain object properties are changed in well thought-out proportions to achieve a certain effect. One of the important principles is timing, or the speed of an action. Similarly, in the domain of automated vehicles, empirical research found that there is a need for balanced acceleration-deceleration patterns for communication with pedestrians [7, 8].

If mobile robots should also behave according to principles of motion design, we hypothesize correlations between certain robot characteristics such as appearance, size, or color with human expectations of their movement. Piwek et al. (2014) [9] tested the uncanny valley of motion hypothesis. They could not confirm a direct link between the human-likeness of a character with an according degree of motion quality and the affinity felt by humans. However, the human similarity approach is rather on a high level, including many object features. In contrast to this, we investigate the effect of the variation of single object features on the expected movement.

## 1.1 Perception of moving objects

In this chapter, we take a differentiated look at the perception of moving objects with regard to effects on paths, distances, and the viewpoints. Since it offers great experimental variation possibilities, we focus on considerations on the research tool virtual reality (VR). Furthermore, we summarize and review common size-speed perception effects.

**1.1.1 Perception of path and distance.**   Depth perception is the ability to perceive the volume of objects as well as their relative position in three-dimensional space [10]. In order to investigate this capability, VR is an increasingly popular tool for evaluation in the domain of moving vehicles. VR makes it easy to change the size and shape of objects and to vary their movement without imposing risks on participants. However [11, 12], found that underestimating distances seems to be a problem in VR, especially at distances greater than 1.0 m. In a range between 2.0 m and 7.0 m [13] report a fairly constant degree of underestimation. In [11], low translation gains and thus no walking at the moment of query promoted accurate distance perception with low estimation errors in a virtual environment. In addition, the authors suggest letting the participants walk at the beginning of an experiment to promote accuracy in the further course of the study.

**1.1.2 The role of viewpoint on the perception of motion.**   Depending on the viewing angle, the perception of motion is promoted by dilation or contraction of boundaries of an obstacle [14]. A dilating image can be seen when facing a robot's motion from the front. When an object approaches, the edges of this object seem to move further away from each other [15].

In the automotive domain [14] searched for an explanation of movement predictions such as time-to-collision that is determined by few parameters. This work suggests that a driver can calculate time-to-collision based on the visual angle and its adjustment over time. However, this human ability remains limited to small visual angles and constant speed of the approaching object [16]. The rate at which the angle of view between any two points of an obstacle changes as it approaches is called $\tau$. It is questionable whether $\tau$ can be generalized for any speed or distance assessments [17, 18]. For example, in real scenarios, adjusting the angle of view is always associated with a change in the distance to other objects that can be used as cues [19]. Even in laboratory experiments, where researchers changed the visual angle without manipulating the distance or speed to an object, the participants used environmental cues to estimate time-to-collision rather than judging the rate of dilating visual angles [20, 21]. It follows that $\tau$ is most likely involved in motion detection, but rather as one source of information among others [22, 23].

Likewise, an intended effect of motion design is always dependent on the abilities of an observer to view the entire path. Only when viewed from above or from the side is the entire path of a linear movement perceptible. From other points of view the movement appears distorted. Nikolaidis et al. (2016) [2] therefore optimized robot motion planners for viewpoint-based legibility.

Acquainting to an object also plays an important role for movement estimations. Without prior knowledge, an object appears to be changed in size related to the distance to the observer and in shape in relation to the viewing angle. When the observer has an accurate mental model of the object, it can be perceived with constant size and shape despite changes of its location [24].

**1.1.3 Perception of speed depending on the size of objects.**   The comparison of the speed perception of large moving objects with smaller moving objects revealed the occurrence of a size-speed illusion. In size-speed illusions a large object seems to be moving slower than a small object traveling at the same speed [19, 25]. This means, in practice observers underestimate the speed of larger objects like airplanes or trains.

The opposite phenomenon is the size-arrival effect. When observers are asked to judge the time it takes an approaching object to arrive at a predefined position, they tend to provide lower estimates for larger objects, suggesting that their speed might be perceived as higher [26, 27].

There are some proposals on the factors that contribute to these contradictory effects. For example, size-arrival considerations usually imply a frontal view of the target and movement is along the depth axis. Hence, dilating images have to be used by the observer to judge speed, which is more difficult with small vehicles. Another study showed that the eye movement behavior of the participants was different when assessing the speed of small and large objects [28]. The eye fixations were located further away from the front of the large objects. In a follow-up study the size-speed illusion was eliminated by placing small dots on the front of the small and large objects to entice participants to fixate their eyes on the dots [28]. The speed was then perceived equally.

Hence, varying viewpoints and fixation points may contribute to the confusion around speed-arrival bias and size-speed illusion. We created Table 1 to illustrate the alignment of size-speed effects with experimental conditions.

In computer simulations by [29], participants approached a floating object and tried to jump over it without collision. The participants jumped significantly later over small objects than over larger objects. This indicates that small objects are perceived more slowly. In [19], participants had to judge which of the two vehicles, train or car, were perceived as being faster in approaching an intersection. The participants significantly underestimated the speed of the train, compared to the car. Clark et al. (2016) [28] varied the distance of the observers and

**Table 1. Review in the field of size-speed effects.**

| Reference | Viewpoint | Size variation | Initial distance of the vehicle from the observer | Effect on the perception of speed |
|---|---|---|---|---|
| Delucia et al. (1999) [29] | Motion in depth axis straight towards the observer | Floating squares: 4.85 m x 4.85 m, 21.82 m x 21.82 m, 38.78 m x 38.78 m | 387.84 m | Small objects are perceived as slower than large objects. |
| Clark et al. (2013) [19] | Lateral approach with viewpoint change from the depth axis to the side | Car: 3.80 m (l), 1.80 m (w), 1.45 m (h); train: 209 m (l), 2.20 m (w), 3.15 m (h) | 200 m, 100 m, and 60 m, observer location 6 m from the vehicle's path | Underestimation of the speed of the train compared to the car. |
| Clark et al. (2016) [28] | Motion in depth axis towards the observer | Car, train | 36 m, 18 m | Large objects are perceived as slower than small objects. |
| Petzold et al. (2016) [26] | Lateral approach with viewpoint change from the depth axis to the side | Car, truck, train | Far condition: 100 m (75 m at end of the video); near condition: 60 m (15 m at the end of the video) | The train and the truck are perceived as slower than the car. |
| Beggiato et al. (2017) [27] | Lateral approach with viewpoint change from the depth axis to the side | Car (Smart), truck (Iveco IV 35S15) | Distance to the vehicle not reported; observer was 0.5 m from the roadside | The truck seems to arrive earlier, which indicates a higher perceived speed. |
| Ackermann et al. (2019) [30] | Lateral approach with viewpoint change from the depth axis to the side | Car (Smart), car (BMW I3), truck (Mercedes-Benz Van) | Not reported | Large vehicles provoke the slowest reaction time, indicating that large objects are perceived as slower than small objects. |

repeatedly confirmed the underestimation of the speed of larger vehicles. Speed and time-to-arrival judgments were tested for virtual depictions of a train, a truck, and a car in [26]. The speed was underestimated for the train and the truck compared to the car. Time-to-arrival was overestimated for the truck compared to the car, and no effect was found for a train. Beggiato et al. (2017) [27] assessed the effects of vehicle size, speed, and the participant's age on expected braking initiation. Vehicle size affects time-to-arrival estimations, but not gap acceptance. A larger vehicle appears to arrive earlier, indicating a higher perceived speed. Ackermann et al. (2019) [30] evaluated the reaction times of participants to approaching vehicles of different sizes and with different deceleration profiles. They found a significant effect of vehicle size on reaction time, namely that large vehicles lead to the slowest reaction times, indicating that larger objects are perceived more slowly than smaller ones. The vehicles were two small passenger cars and one small truck, of which the two small ones were probably perceived to be the same size.

## 1.2 Designing motion for legibility

Intentions can be derived from movement patterns even when no human forms are depicted [31]. Heider et al. (1944) [32] showed this can be done with simple objects, like two-dimensional triangles. Based on such findings, the design for legible motion is applied in various research areas, such as automated vehicles and robotics.

Fuest et al. (2018) [8] let participants drive a car with the task to either perform a maneuver that conveyed the intention of yielding priority to pedestrians or claiming their own right of way. The authors combined the resulting driving profiles to form an average trajectory [8]. Ackermann et al. (2019) [30] used vehicle deceleration as an informal communication cue in their study. The authors varied specific parameters, deceleration rate, onset of deceleration, vehicle size, and speed, in two experimental, video-based simulations in regard to detection of yielding intentions. They demand communication between automated vehicles and pedestrians to be optimized by applying smooth and early deceleration that incorporate vehicle speed and size.

For automatic control of robots [1], developed a formalism to mathematically define legibility of motion on the basis of optimizing cost. However, the developed framework does not provide insights into the motion parameters that are responsible for legibility. Knight et al. (2014)

[33] applied the Laban Effort System on robot motion. The Laban Effort System is an ontology for the dynamics of how humans generate expressive motion in the discipline of acting/theater. It is defined by the categories time, weight, space, and flow. The combination of efforts displayed during a path (e.g., acceleration, focus) can indicate something about an agent's inner state (e.g., confidence). Participants had to move an object between two points and thereby evoke different states (happy, sad, confident, shy, rushed). In limiting the implementation to three degrees of freedom the authors could discover principles that connect movements to communication of the robot's state. They found that the clearest communication of a shy state involves direct paths with hesitations and that timing characteristics are one of the most significant features separating the different paths.

Due to its origin in human acting, the Laban Effort System is most applicable for humanoid robots. Additionally, most of the aforementioned works have in common that the description of the movement is not broken down into single parameters. Therefore, a transfer to other shapes of objects and applying the movement in new settings is difficult. When we conducted a human-robot spatial interaction analysis, we found a link between human attention processes and robotic movement parameters like the path and the execution time [5]. This suggests that detailed investigations of the parameters that constitute movements are necessary in order to be able to design movement to convey an intention.

## 1.3 Hypotheses

A human-inspired back-off has been a viable method to communicate yielding priority to humans in earlier works [3–5]. To fulfill the purpose of legibility for observers, a back-off movement has to be executed with a specific path and speed to be perceivable and expressive. On the other hand, to be efficient and executable (time-loss, obstacles), evasive maneuvers should be minimized. To allow the application in a variety of autonomous moving systems, the main research aim of this article is to investigate how appropriate back-off design can be modeled and translated to different sized robots and observer's viewpoints.

Therefore, we investigate how humans adjust the motion parameter "back-off length" to implement a back-off that sufficiently communicates the intention of yielding priority depending on the size of the robot and the viewpoint. We expect that bigger robots have to move further away from the participant to be expressive because movements have to be scaled in relation to object size to support their correct perception.

*H*1: The required back-off length increases with increasing robot size.

Viewing the same movement from different perspectives leads to different perceived paths. Therefore, we expect that the chosen back-off length is affected by the viewpoint.

*H*2: The chosen back-off length differs between the two viewpoints "frontal view" and "lateral view".

The literature review showed that perception and expectation of speeds of moving objects is affected by their size and the viewpoint. Hence, we investigate how participants choose the parameter "back-off speed" for communicating the intention of yielding priority. The presence of size-speed effects allows hypothesizing that perception and design of back-off speed is also a matter of the robot's size. Earlier studies suggest that the speed of larger objects is underestimated in lateral approaches. We expect that participants will compensate this effect by applying higher speeds for the larger robots in the lateral view of a back-off.

*H*3: The preferred back-off speed increases with increasing robot size in the lateral view.

Seen from the front, the size-arrival effect seems to have a greater impact. In this case, the speed of smaller objects is underestimated. We expect that the participants will compensate this effect by applying lower speeds for the larger robots in the frontal view of a back-off.

*H*4: The preferred back-off speed decreases with increasing robot size in the frontal view.

## 2 Methodology

Due to the possibility to vary all factors in small increments, we conducted an experiment in VR. In contrast to our previous experiment where the participants were exposed to two standard versions of a back-off movement [5], they were involved in the adjustment of the back-off parameters in this experiment's co-creation approach [34].

Chapter 2.1 describes the VR environment, planned fixed effects, random effects and measures. Chapter 2.2 explains the instructions and sequence of the experiment. As supporting information an overall system scheme block diagram is provided (S2 Fig). The study design and procedure has been approved by the ethics commission of the Technical University of Munich under number: 91/19 S-SR.

### 2.1 Study design

#### 2.1.1 Virtual environment and technical configuration.   The study environment shows a corridor crossing with a door on one side (Fig 1). Participants could move freely in this

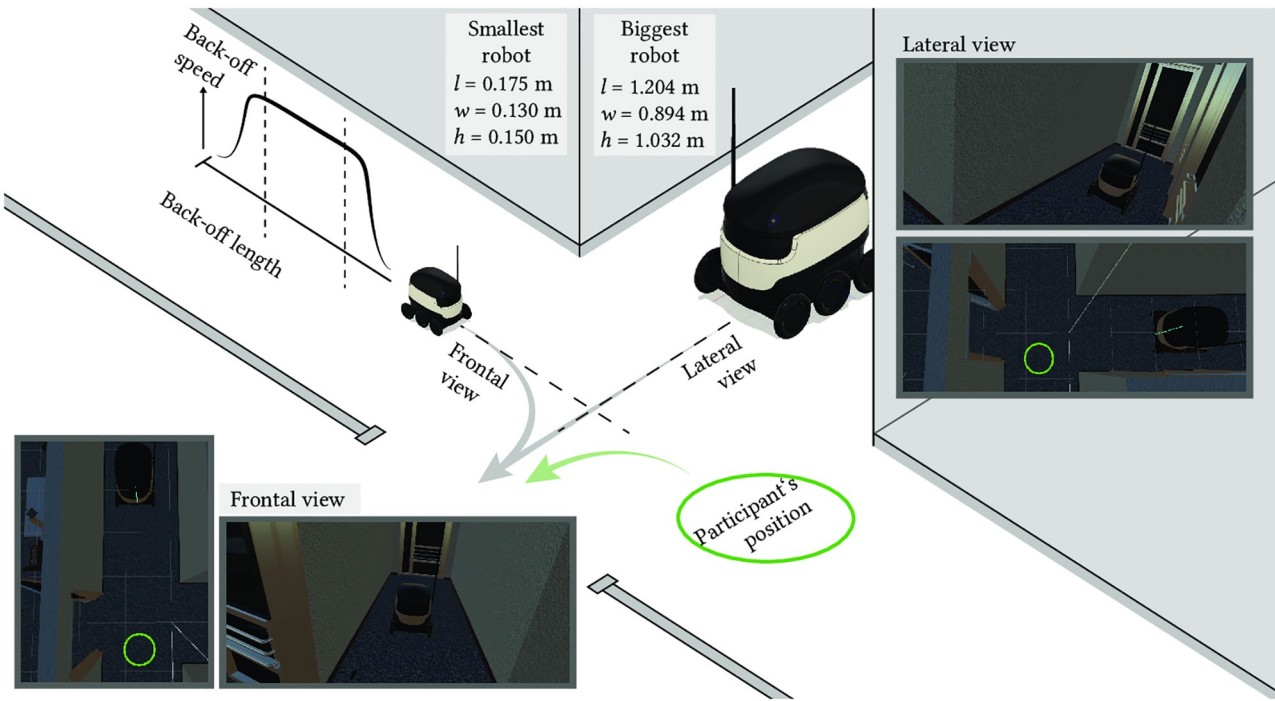

**Fig 1. Illustration of the virtual study environment, the experimental conditions, and measures.** Robots with different sizes ranging from 0.175 m x 0.130 m x 0.150 m to 1.204 m x 0.894 m x 1.032 m (length x width x height) were manipulated from two viewpoints, the frontal view and the lateral view. In a training phase, participant and robot moved through a door and the robot performed a predefined back-off (arrows represent approximations to the planned paths). In the design phase of the experiment, participants adjusted the back-off length and back-off speed of the robot from a stationary position (participant's position).

office-like setting except for parts of the study when they were asked to stay in one location. Participants wore a head-mounted display (HTC Vive Pro) with a dual AMOLED screen with a resolution of 1080 x 1200 pixels per eye and a field of view of 110 degrees. During the experiment, participants held an HTC Vive controller in their hands to enter and confirm the back-off length and *speed*. Whenever a person approached the border of the experimental area by less than 0.5 m distance, a blue mesh was displayed to signal a possible collision with an obstacle in the real space. The environment was programmed in Unity (Version 2018.2.17f1) including the SteamVR Plugin (Version v2.2.0).

**2.1.2 Fixed and random effects.** The robot model is based on the delivery robot manufactured by Starship Technologies and designed with the software Blender. It resembles the original Starship robot in its proportions. Fifty robots were designed in various sizes, the smallest being 0.175 m x 0.130 m x 0.150 m, the largest being 1.204 m x 0.894 m x 1.032 m (length x width x height). The remaining robots were linear scaled versions between the minimum and maximum robot size.

We let the robot approach from two directions (Fig 1). One approach direction was straight toward the participant, resulting in a frontal view. The other direction was perpendicular to the original orientation of the participants, which led to a lateral approach of the robot relative to the human observer.

Since a participant was exposed to robots of different sizes, the assumption of independence from errors was violated. In this case, individual differences and sequential effects can lead to individually subjective reactions and preferences by the participants. Therefore, we considered the participants' preference as a random effect. Furthermore, individual physique and the resulting differing eye-points of the participants led to varying perspectives. In order to take this geometric effect into account, we proposed the height of the participants as an additional random effect.

**2.1.3 Measures and robot behavior.** A participant adjusted the back-off length via holding down the trigger button on the HTC Vive controller. During the adjustment, the robot was displayed transparently and overlaid with the end position of the approached robot. The superimposed adjustable robot had to be placed at the desired back-off end position. The trigger button was released for this purpose. The back-off length could be varied between 0 m and 4.5 m, which corresponds to the end of the corridor. Back-off speed could be varied from 0.1 to 2.0 m/s in increments of 0.1 m/s using the HTC Vive controller's control dial. This represents the preferred maximum speed during the maneuver. A comparable approach can be found in [35], where the participants designed the behavior by controlling the robot a wireless keyboard. Also in [36], the preference or dislike for certain robot behaviors was recorded at the push of a button when participants defined the minimum accepted frontal and lateral distances between themselves and a robot when passing through aisles.

Accordingly, a back-off maneuver is calculated based on the input of the two parameters set by a participant, length to cover and maximum speed (Eq 1). The execution of speed as a function of time is associated with a certain amount of acceleration and deceleration during the maneuver. In order to create a realistic appearance of acceleration and deceleration, the speed is modeled with a cosine function. In the first term the negative sign is used for the acceleration phase of the robot and the positive sign for the deceleration phase. *BO.speed* represents the maximum speed during the backwards movement, and *BO.length* the back-off length. The cosine function is adjusted by dividing *BO.speed* by *BO.length*. This adapts the acceleration to back-offs with a large difference between speed and length. The variable *time* always starts at 0

s when accelerating or braking during the back-off maneuver.

$$v(t) = \pm\ 0.5 \cdot BO.speed \cdot \cos\left(\frac{BO.speed}{BO.length} \cdot \pi \cdot time\right) + 0.5 \cdot BO.speed. \qquad (1)$$

At the end of the experiment the participants answered a questionnaire. In addition to demographic data on the age and height of the participants, general questions regarding the back-off were asked. The participants' experiences with VR environments were assessed on a five-point Likert-scale from "no experience at all" to "regular usage". The subjective comprehensibility of the back-off movement was determined on a five-point Likert-scale from "the back-off is illegible" to "the back-off is very legible".

The random distribution of robot sizes in the course of the experiment (Chapter 2.2) brings to light the issue that some participants are presented with two consecutive robots that are very similar in size. In order to determine whether these size differences can be distinguished in VR, the participants were interviewed during the experiment at each change to a new robot size whether there was a difference to the previous model that was greater, smaller, or if they were the same size.

## 2.2 Procedure

The participants signed the informed consent paper, that explains the research question, general procedure, the risks when using VR devices and data protection. The consent form was approved by the ethical commission. They then put on the head-mounted display and familiarized themselves with walking in the virtual environment. The laboratory offers a 4.00 m x 5.00 m area for the participants' movement in VR. The experiment consisted of two parts, a "training phase", and a "design phase".

As a training phase, the back-off was demonstrated to the participants to give them a first impression of its execution and communicative purpose. Therefore, a first encounter with the robot was arranged, during which the participants could move freely. Starting at the end of the virtual hallway, participants were given the task of walking through the door to their left (Fig 1). The robot started in the straight alley and also moved toward the door. The robot's movement was synchronized in time with the participant, so that both arrived at the door at about the same time. Instead of moving further through the door, the robot performed a back-off with a predefined back-off length of 1.2 m, inspired by the dimensions of a personal space [37]. The maximum speed of the predefined movement was set to 1.4 m/s. This is an average comfortable walking speed of pedestrians between 20 and 40 years [38]. After the back-off, the participants could enter the room on the left side. In addition, the intention of the back-off was explained to the participants. This part could be repeated if the participants wished to experience the back-off movement again.

In the design phase, the participants stayed in one place. Their initial body and head orientation were along the straight alley. Each participant experienced five different robots (Fig 2). For this purpose, the 50 robots of different sizes were divided into five clusters. One robot was selected from each cluster for each participant. The selection of a robot within a cluster was randomized, and the robot of the respective size was not available to subsequent participants until all robots had been selected. The order of occurrence of the clusters was also randomized.

Each participant adjusted the back-off for both viewpoints in a sequential order, first the frontal view, then the lateral view. To initiate the back-off design according to Chapter 2.1.3, a robot always started form the end of the respective alley and drove toward the door, then came to a standstill. Subsequently, back-off length and back-off speed were adjusted simultaneously. The participants were advised to enter the required minimum back-off length, that is as small

**Fig 2. Robot selection during the back-off design phase of the experiment.** For each participant, one robot is selected from each of five size clusters in a randomized order. This robot will not be available to subsequent participants until all robots have been selected. The order of occurrence of size clusters is also randomized for each participant.

as possible but as large as necessary to be expressive. No additional advice was given for the back-off speed. The back-off with the selected settings was shown in a repeating animation. During this replay, the back-off speed could be adjusted again. If necessary, also the complete back-off setting could be revised as often as required until the participant confirmed the selection for the respective robot and viewpoint. After having applied settings for five robots the questionnaire was answered (Chapter 2.1.3).

Every ten participants, all robot models were used once and the selection procedure started again with the subsequent participant. The whole procedure took 30 minutes per participant. Each participant was introduced to the control (2 minutes) and completed the training phase (3 minutes). The back-off design with five robots from two viewpoints was carried out in approximately 20 minutes. Finally, five minutes were spent on the questionnaire.

## 2.3 Sample

In total, 50 participants took part in this study, ranging from 19 to 59 years, with a mean age of $M = 25.50$ years ($SD = 6.55$ years) and a mean body height of $M = 1.725$ m, ($SD = 0.099$ m). The volunteers were recruited on the campus of the Technical University of Munich and received no payment. The participants reported on average $M = 2.34$ ($SD = 0.97$) experience in virtual environments, and reported that the back-off is legible to them with a mean value of $M = 4.48$ ($SD = 0.67$).

## 3 Statistical analysis and results

We analyzed the effects of robot size and viewpoint on back-off length and back-off speed using a linear-mixed-effects-model. With the application of a linear-mixed-effects-model it is possible to consider longitudinal effects that occur during the course of the experiment, such as learning effects due to trial repetitions and participant-specific variability. For this we use the "nlme" package in the R environment. A top-down strategy similar to [39, 40] was applied to create the model for the given data. First, the fixed effects were added to the model as a well-specified mean structure. Random effects were subsequently added. A comparison of the more detailed models including the effect in question (full model) with the model without the effect (null model) shows whether the effect in question has a significant contribution to the model's quality. For this purpose an ANOVA was performed. The threshold for statistical significance is set to $\alpha = 0.05$. The aforementioned comparison was iterated several times to test for interaction effects, random intercepts, and slopes of the fixed and random effects (Chapter 2.1.2). We choose the length of the robot as the modeling parameter that represents the fixed effect "robot size".

**Table 2. Model selection for back-off length using information criteria AIC, BIC, and p-value between model (5) and the models (1)–(4).**

| No. | Model[a] | AIC | BIC | *p* |
|-----|----------|-----|-----|-----|
| (1) | *BO.length ∼ robot.length + viewpoint, 1\|participant* | 41.79637 | 62.86941 | <0.001 |
| (2) | *BO.length ∼ robot.length * viewpoint, 1\|participant* | 39.23269 | 64.52034 | <0.001 |
| (3) | *BO.length ∼ robot.length * viewpoint, 1\|participant.height* | 208.75025 | 234.03790 | <0.001 |
| (4) | *BO.length ∼ robot.length * viewpoint, viewpoint\|participant* | 43.23269 | 76.94955 | - |
| (5) | *BO.length ∼ robot.length * viewpoint, robot.length\|participant* | -35.10278 | -1.385913 | |

[a]Notation used is in R-format.

## 3.1 Back-off length

In this chapter, we describe the process of model generation for the required back-off length. All models are summarized in Table 2.

The first model includes main effects for robot length and viewpoint and no interaction effect (Table 2, model (1)). Additionally, we implement random intercept for each participant, but no random slope. The random intercept applies to both fixed effects. Next, a model (2) with interaction effect of main effects is compared to the null model. The comparison shows a significant difference between these two models (*p* = 0.0327). As it improves the model quality, it is advisable to consider the interaction effect. A further model (3) is created, which includes the body height of the participants as a random effect instead of the participant number. This model shows relatively high AIC and BIC values. The factor is therefore considered probabilistic and random. This allows us to neglect the body height in the following. Model (4) is set up to investigate whether random slopes for the viewpoint improve the quality of the model. The comparison with model (2) shows no significant difference between the models. Hence, there is no significant effect of a random slope for each viewpoint per participant on the model quality. The modeling of a random slope for the viewpoints can therefore be neglected. Additionally, we investigate whether random slopes for robot length improve model fitting (5). The comparison between model (5) and (2) shows a significant difference (*p* < 0.001). This means that the inclusion of random slopes for increasing robot length per participant has a significant influence on the model quality.

The model (5) that is finally chosen yields the lowest AIC and BIC values and differs significantly from all other model representations. Both main factors involved contribute significantly (Table 3). The resulting linear equation, which predicts the required back-off length for different robot sizes and viewpoints accordingly formulates to Eq 2. For the variable *viewpoint* "0" must be applied for the lateral and "1" for the frontal view. i = [1, . . ., n] represents the participants and accounts for random intercept and slope. $\varepsilon$ represents the probabilistic random

**Table 3. Results of the linear mixed model (5) for predicted back-off length.**

| Random effects | Name | | SD | | |
|----------------|------|---|-----|---|---|
| (intercept) | $c_i$ | | 0.3016096 | | |
| robot.length | $a_i$ | | 0.3626634 | | |
| **Fixed effects** | | **Estimate** | **Std. error** | *t* | *p* |
| (intercept) | $c$ | 0.5063625 | 0.05196386 | 9.744512 | <0.001 |
| robot.length | $a$ | 0.1997319 | 0.06466969 | 3.088494 | 0.0021 |
| viewpoint | $b$ | -0.1042546 | 0.04126442 | -2.526501 | 0.0119 |
| robot.length x viewpoint | $d$ | 0.1381270 | 0.05478846 | 2.521097 | 0.0120 |

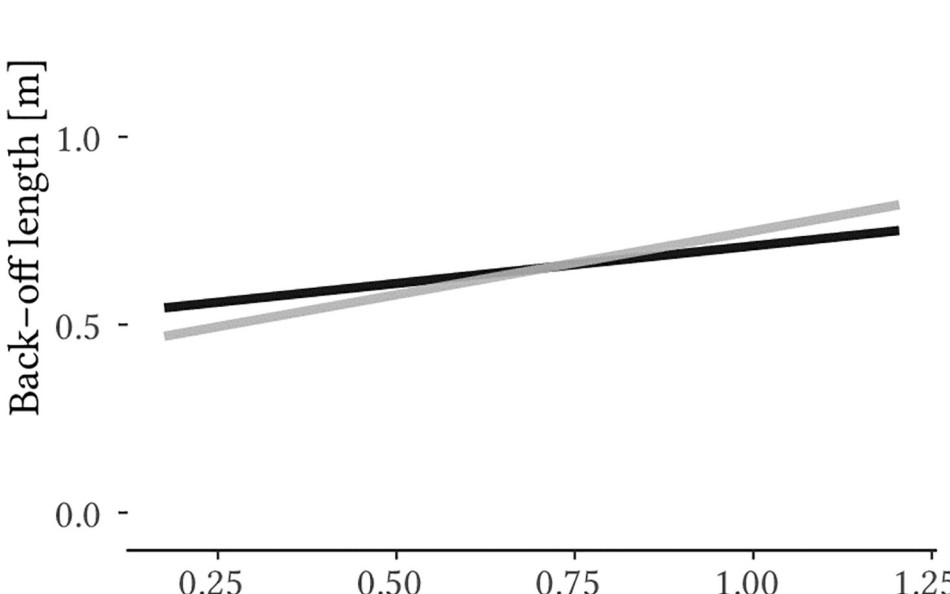

**Fig 3. Predicted necessary back-off length depending on robot size and viewpoint.** The black line represents the linear equation for the frontal view, the gray line represents the linear equation for the lateral view. The number of measurements used to create the model is $N = 500$.

error term. A visual representation of the model is shown in Fig 3.

$$BO.length = c_i + a_i \cdot robot.length + b \cdot viewpoint + d \cdot robot.length \cdot viewp. + \varepsilon. \quad (2)$$

## 3.2 Back-off speed

This section considers the effects on the preferred back-off speed. The process of model generation is described and the models are summarized in Table 4.

Model (1) contains the main effects for robot length and viewpoint, random intercept and no random slopes. A model that considers the interaction effect of main effects (2) is compared to the null model. The comparison does not show a significant difference between the two

**Table 4. Model selection for back-off speed using information criteria AIC, BIC, and p-value between model (5) and the models (1)–(4).**

| No. | Model[a] | AIC | BIC | p |
|---|---|---|---|---|
| (1) | $BO.speed \sim robot.length + viewpoint, 1|participant$ | 156.1905 | 177.2636 | $< 0.001$ |
| (2) | $BO.speed \sim robot.length * viewpoint, 1|participant$ | 157.5327 | 182.8203 | $< 0.001$ |
| (3) | $BO.speed \sim robot.length + viewpoint, 1|participant.height$ | 441.2020 | 462.2750 | $< 0.001$ |
| (4) | $BO.speed \sim robot.length + viewpoint, viewpoint|participant$ | 160.1905 | 189.6928 | - |
| (5) | $BO.speed \sim robot.length + viewpoint, robot.length|participant$ | -29.22402 | 0.27823 | |

[a]Notation used is in R-format.

**Table 5. Results of the linear mixed model (5) for predicted back-off speed.**

| Random effects | Name | | SD | | |
|---|---|---|---|---|---|
| (intercept) | $c_i$ | | 0.5462427 | | |
| robot.length | $a_i$ | | 0.5212022 | | |
| **Fixed effects** | | **Estimate** | **Std. error** | **t** | **p** |
| (intercept) | $c$ | 1.2991656 | 0.08023990 | 16.191017 | < 0.001 |
| robot.length | $a$ | -0.2554644 | 0.07830956 | -3.262238 | 0.0012 |
| viewpoint | $b$ | -0.0184000 | 0.01530123 | -1.202517 | 0.2298 |

models (1) and (2). The consideration of the interaction effect therefore does not contribute to model quality and can be neglected. A further model (3) is created, which includes the body height of the participants as a random effect instead of the participant number. This model shows relatively high AIC and BIC values and can be considered probabilistic and random. As in Chapter 3.1 the body height is neglected. Model (4) is set up to investigate whether random slopes for the viewpoint improve the quality of the model. The comparison with model (1) shows no significant difference between the models. Consequently, there is no significant effect of a random slope for each viewpoint per participant on the model quality. The modeling of a random slope for the viewpoints can therefore be neglected. Additionally, we investigate whether random slopes for robot length improve model fitting (5). The comparison between model (5) and (1) shows a significant difference ($p < 0.001$). Therefore, the inclusion of random slopes for increasing robot length per participant has a significant influence on the model quality.

Model (5) provides the most accurate fit. It yields the lowest AIC and BIC values and differs significantly from all other model representations. Of the two main factors, only robot length contributes significantly (Table 5). Since it was conceived as a fixed effect in the study, the viewpoint was nevertheless left in the model. The resulting linear equation, that predicts the back-off speed for different robot sizes and viewpoints accordingly formulates to Eq 3. For the variable *viewpoint* "0" must be applied for the lateral and "1" for the frontal view. i = [1, . . ., n] represents the participants and accounts for random intercept and slope. $\varepsilon$ represents the probabilistic random error term. A visual representation of the model is provided in Fig 4.

$$BO.speed = c_i + a_i \cdot robot.length + b \cdot viewpoint + \varepsilon. \tag{3}$$

## 3.3 Back-off time

The execution time of a movement is a parameter that can be used to investigate speed and distance traveled in combination. If a participant changes the required back-off length and leaves the desired back-off speed at the same value, it means that with this setting he or she also changes the execution time of the maneuver. Similarly, if a participant changes the desired back-off speed while leaving the same required back-off length, he or she also changes the execution time. However, back-off length and back-off speed show a weak positive correlation in the dataset ($r = 0.36$, $p < 0.001$) (S1 Fig), which indicates a tendency that participants combined greater back-off lengths with higher speeds. It is possible that participants wanted to compensate greater back-off length with an increase in speed to keep the execution time at a similar level. Thus, the investigation of length and speed in combination may lead to a better understanding of the perception and expectations of motion. For this purpose we calculate an additional parameter composed by the speed of driving backwards over a certain length, the back-off time, in an exploratory analysis.

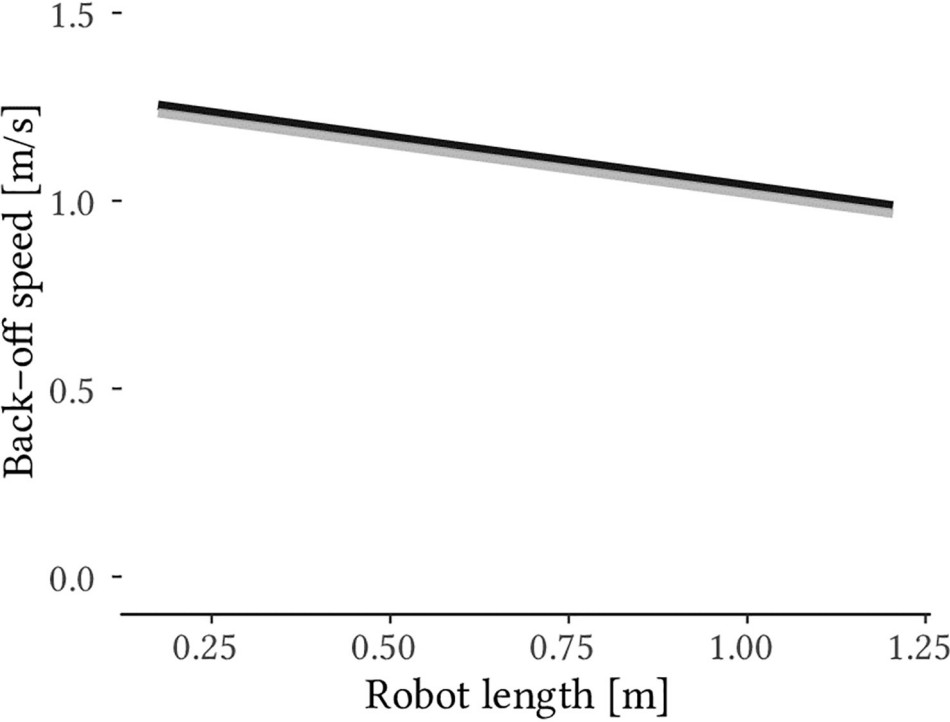

**Fig 4. Predicted preferred back-off speed depending on robot size and viewpoint.** The black line represents the linear equation for the frontal view, the gray line represents the linear equation for the lateral view. The number of measurements used to create the model is $N = 500$.

The back-offs designed by the participant were recorded with time stamps of 0.1 s. It is possible to extract the back-off time of each back-off movement from these records (Fig 5). The back-off time is the total time that the back-off maneuver takes from the initiation of acceleration to the renewed standstill according to the behavior designed in Chapter 2.1.3.

The mean back-off time of the data set is $M = 1.042$ s ($SD = 0.5229$ s) and the median is $Mdn = 1.000$ s. A Shapiro-Wilk normality test shows that the data set has a right skew and is not normally distributed ($W = 0.90913$, $p < 0.001$, $Skewness = 1.45$). The number of measurements is reduced to $N = 478$ for this analysis due to mislabeled recordings of 20 trials that cannot be used and missing data of two trials.

## 4 Discussion

In the collected data we find an increase of the required back-off length with increasing robot size. This confirms our expectations and we therefore accept Hypothesis 1. Additionally, the application of two different viewpoints has an effect on the selected back-off length. Even if the effect is small, we accept Hypothesis 2. The analysis of the selected back-off speed does not correspond to our hypothesis. The viewpoint does not contribute significantly in the linear-mixed-model. This leads to the conclusion that there is neither a significant increase of the preferred back-off speed with increasing robot size in the lateral view nor a decreasing back-off speed with increasing robot size in the frontal view. We therefore reject Hypotheses 3 and 4.

In this chapter we discuss the application of the results and how they relate to the state of the art. Furthermore, we discuss the means by which the experiment was performed, including the limitations, and what we recommend to investigate in subsequent experiments.

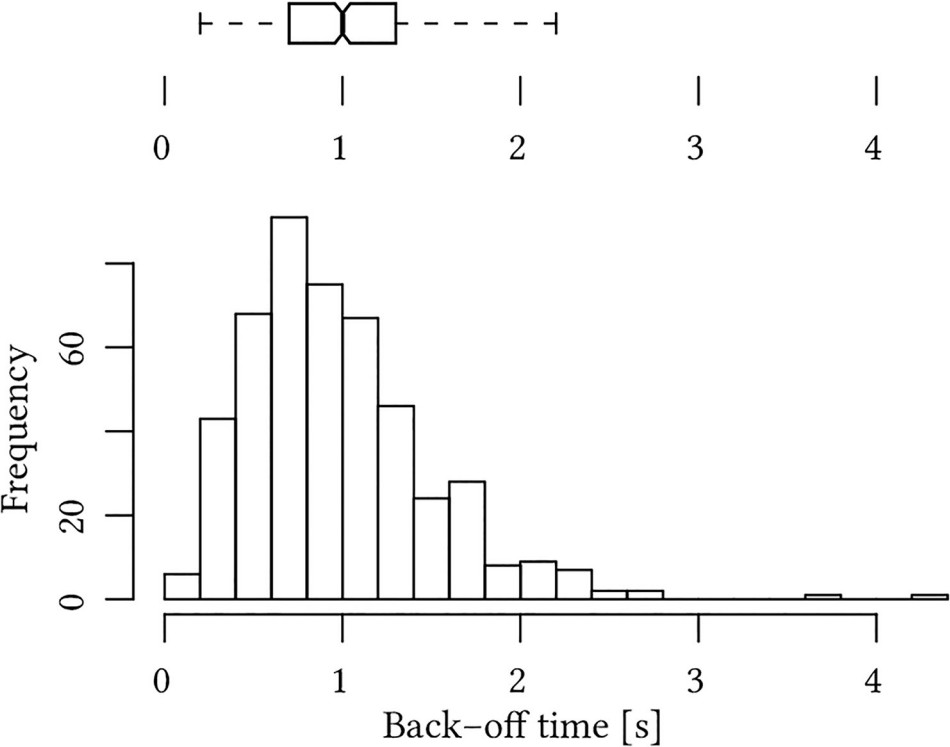

**Fig 5. Histogram and boxplot of the back-off time.** The cumulative number of designed back-offs is displayed in clusters with a width of 0.2 s back-off time. The boxplot displays median, quartiles, whiskers and 95% confidence intervals. The number of measurements is $N = 478$.

## 4.1 Interpretation

A calculation example demonstrates the application of the results from the presented experiment. We use the model to predict required back-off length with robots of larger sizes (Eq 2) and insert the parameters defined by the model (Table 3). This could result, in the following recommendation for the use of two new differently sized robots with a tailored behavior for interaction with a pedestrian from the lateral view: A robot, manufactured with double the size (0.5 m length increased to 1 m length) would need a back-off length increase from 0.606 m to 0.706 m to be legible. This is equivalent to an increase of 16.6%. Assuming that this approach would prove valid in reality, a remarkable implication of the model is that the required increase is not proportional. While the size of a robot is doubled, the modeled approach predicts that the larger robot would need only a fraction of this increase in movement capabilities.

The viewpoint has a small effect on the back-off length in the presented experiment. A possible explanation is the proximity between human and robot regarding the movement. The lateral view differs from the lateral views in other studies. In our investigation, the distance between the observer and the lateral movement axis of the robot was approximately 1 m. The distance depended on the positioning of the participant within the scope of minor deviations and the size of the robot. In [19], the train and the car were further away along the axis of motion and the observer was at a greater distance from the axis of motion. A larger change in the angle of view compared to our experiment can therefore be expected.

In general, we do not find strong effects regarding the influence of different robot sizes on the back-off speed. In particular, it does not seem plausible to compare the results to the

examined effects on size-speed biases. The presented experiment is not only about the perception of movement, but also about its design. A participant experienced a continuous feedback loop including the variation of movement and the perception of change. This prolonged exposure to a variation in movement may have mitigated the effects. Moreover, the back-off was performed away from the viewer, whereas in the literature the movement is always performed toward the viewer. The research base for movements away from an observer compared to movements toward an observer is small, and it is yet unclear what role size-speed illusions play in such maneuvers.

The sample, primarily comprised of young university students, showed a high degree of comprehensibility of the back-off movement as indicated by the questionnaire (cf. Chapter 2.3). Regarding this characteristic, the sample is similar to the sample in our previous experiment [5]. Therefore, we interpret the results of both experiments in comparison.

In the recorded data set the mean required back-off length is $M = 0.641$ m ($SD = 0.336$ m) ($Mdn = 0.566$ m). This is in the order of magnitude of the comparatively long back-off movement from our previous experiment, which had a back-off length of 0.54 m [5]. However, the long back-off movement was less efficient for human-side interpretation. In contrast, the resulting back-off time $M = 1.043$ s ($SD = 0.523$ s) ($Mdn = 1.000$ s) is in the range of the more efficient short back-off movement from our previous experiment, which had a back-off time of 1 s [5].

The contradiction between these two comparisons could be an indication that considering the duration of a movement is a better indicator of its expressiveness. This is in agreement with the conclusions of [6] who regard the timing of an action as one of the important principles in motion design. Also, in the studies of [33] timing is one of the most important features separating the different paths.

## 4.2 Limitations

The way humans interact with objects in a virtual environment differs from real physical interaction. In this context [11], found that the method for measuring distance has a significant influence on the estimation error. Our methodology is different compared to the methods used in [11]. The participants moved the robot to the desired location using the HTC Vive controller. They received constant feedback on the implemented distance, as a virtual robot was faded into the environment. It can be argued that the superimposed robot minimized the estimation error. In general, the underestimation of distances in VR is a current field of research. However, the effect is not necessarily limited to virtual environments. [41] found such an underestimation also in physical environments.

Another aspect to be validated in the presented experiment is the perception of the robots' size. Differences in robot sizes were correctly detected in 96.4% of robot changes. Thus, the advantages of the high degree of experimental control and variability through many robot models that VR offers have merely caused a small number of misinterpretations.

As expected by the findings of [24] the construction of a mental model may have created an expectation on the back-off maneuver from the first encounter. However, the back-off length of 1.2 m in the training phase is far from the mean values designed in the design phase of the experiment. This is an indication that the mental model via the predefined back-off does not prevent the participants from designing according to their own preferences.

For the back-off speed hypothesis it must be taken into account that when setting the speed, the value was preset to 1.4 m/s before the participants could adjust it. The participants may have been influenced by the preset value. In fact, the mean value selected by the participants during the experiment is close to this initial value.

When setting up an office environment that imposes spatial constraints, we conducted the experiment to maximize external validity. Hence, by providing reference values for distances that exist in a real office environment, we restricted the participants' freedom of choice. The maximum applicable back-off lengths limited the customization possibilities in this respect. Limited by the corridor, the maximum adjustable back-off length was 4.5 m. On the other hand, we could have provided an artificial white room application where a participant would be able to adjust the back-off without restrictions. This might have created more scope for creativity in the design variations. We expect a greater variance of the parameters in such an experiment. However, the application of such reference values in a real robot would be questionable.

### 4.3 Future work

The form factor of an object plays a role in the eye movement behavior of an observer. [19] found this when examining trains and cars in comparison. Furthermore, the different eye fixation behavior of these two vehicle types affected the perception of their speed. Differently shaped robots could thus influence the expected back-off. In a recent VR experiment with a robot shaped with a long neck, we found that the long neck caused the participants to fixate their eyes either on the upper or the lower part of the robot. Therefore, in future experiments, shape variants that differ from the rather standard box shape of the robot we used may reveal new perspectives on the connection between object shape and human expectations of its movement.

To validate the parameters predicted by the linear model, larger sized examples, such as cars and trucks mentioned in the literature, or fewer variations and thus more recurring robots, could be studied. Fewer robot sizes and a higher number of repetitions for each robot can reduce variance as participants become more familiar with each robot. Replication of the study with a real robot could also be used for validation.

The results of the presented experiment suggest that execution time might be a suitable design parameter considered for further research. The topic of execution times of expressive movements seems promising, and we recommend further research in this direction.

### 5 Conclusion

In previous experiments, we introduced a back-off as a movement strategy of robots to facilitate the sequence of passage at bottlenecks in the spatial interaction between humans and robots. It is a backward movement along the original trajectory of a robot to convey its intention to yield priority to pedestrians. In this earlier experiment we designed two variations of this movement. One was comprised of a short path and a short duration, the other one had a long path and a long duration. Related works in distance perception, size-speed illusions, and viewpoint-based legibility considerations suggest a relationship between the size of the robot and the observer's perspective on the expected execution of that movement. In order to optimize this back-off movement and make it applicable to a variety of robots, we extended our previous research with a participant experiment (N = 50) in a VR environment. We tried to predict suitable parameters for the design of an expressive movement depending on two main factors. The participants set the minimum required back-off length and the preferred back-off speed. The correlation between the increasingly expected back-off lengths with increasing robot size shows that the expectations for the execution of such a movement differ depending on the robot. However, only the assumptions associated with the parameter "length" of the movement were supported by the data set, and the effects we found are rather weak. The exploratory study of execution time is promising, and a follow-up question could be whether

the movements can be trimmed to an optimized execution time while covering minimal distances. This could be seen as a challenge to the results of [30], who conclude that there is no one-fits-all solution for a perfect maneuver applicable to any kind of speed, deceleration rate, or vehicle size. We hope that our methodological approach can be adapted and applied to design the expressiveness of a variety of autonomously moving systems such as robots, cars, or unmanned aerial vehicles.

## Supporting information

**S1 Fig. Plot of back-off length and speed data.**
(TIF)

**S2 Fig. Overall system scheme block diagram.**
(TIF)

**S1 Data. Dataset prepared for the R environment.**
(ZIP)

**S1 Video. Video example of the experiment.**
(ZIP)

## Acknowledgments

We thank Bao Cao for collecting data during his master thesis and Jakob Peintner for designing the robot model.

## Author Contributions

**Conceptualization:** Jakob Reinhardt.

**Data curation:** Jakob Reinhardt.

**Formal analysis:** Jakob Reinhardt.

**Funding acquisition:** Klaus Bengler.

**Investigation:** Jakob Reinhardt.

**Methodology:** Jakob Reinhardt.

**Project administration:** Jakob Reinhardt, Klaus Bengler.

**Supervision:** Jakob Reinhardt, Klaus Bengler.

**Validation:** Jakob Reinhardt.

**Visualization:** Jakob Reinhardt.

**Writing – original draft:** Jakob Reinhardt.

**Writing – review & editing:** Jakob Reinhardt, Klaus Bengler.

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
