## [Decision Letter · Decision Letter 0]

11 Nov 2020

PONE-D-20-27433

Design of a hesitant movement gesture for mobile robots

PLOS ONE

Dear Dr. Reinhardt,

Thank you for submitting your manuscript to PLOS ONE. After careful consideration, we feel that it has merit but does not fully meet PLOS ONE’s publication criteria as it currently stands. Therefore, we invite you to submit a revised version of the manuscript that addresses the points raised during the review process.

Please try to revise your manuscript and respond to all the reviewers' comments. 

We look forward to receiving your revised manuscript.

Kind regards,

Quan Yuan, Ph.D.

Academic Editor

PLOS ONE

Journal Requirements:

Reviewers' comments:

Reviewer's Responses to Questions

**Comments to the Author**

1. Is the manuscript technically sound, and do the data support the conclusions?

Reviewer #1: Partly

Reviewer #2: Yes

2. Has the statistical analysis been performed appropriately and rigorously? 

Reviewer #1: Yes

Reviewer #2: Yes

3. Have the authors made all data underlying the findings in their manuscript fully available?

Reviewer #1: Yes

Reviewer #2: Yes

4. Is the manuscript presented in an intelligible fashion and written in standard English?

Reviewer #1: Yes

Reviewer #2: Yes

5. Review Comments to the Author

Reviewer #1: This research considered the construction of human similarity of a robot. The method is proposed to make up a robot’s appearance and behavior that is perceived as human-like. However, the presentation and writing of the paper need to be substantially enhanced. Also, some key information and results should be explained in detail.

1. The presentation and writing of the paper must be substantially improved. The writing of current manuscript is hard to follow.

2. The highlight or important contribution of this paper is missing. Author needs to write the contributions of this paper clearly in section ‘Introduction’.

3. The presentation of the methodology in this paper is vague. The framework of section 2 needs to be improved.

4. This paper is too long, and some key information is hard to be found. The reviewer suggests that the paper needs to be condensed and some key information needs to be highlighted.

Reviewer #2: In this paper, a back-off movement strategy is designed to convey the robot intention to others. The study is interesting. There are some problems as follows.

1) Why does this article choose a back-off movement strategy of robots to express the motion intention? Please explain it in details in Chapter 1.3. The innovative points of this article need to be further condensed and summarized.

2) The back-off movement proposed strategy lacks a detailed process of the method, an overall system scheme block diagram and flowchart in Chapter 2.

3) Figure 3 and Figure 4 lack specific curve annotations.

6. PLOS authors have the option to publish the peer review history of their article (what does this mean?). If published, this will include your full peer review and any attached files.

Reviewer #1: No

Reviewer #2: No

---

## [Author Response · Author response to Decision Letter 0]

20 Dec 2020

The response is provided as a seperate file named "Response to reviewers.pdf".

---

## [Decision Letter · Decision Letter 1]

26 Jan 2021

PONE-D-20-27433R1

Design of a hesitant movement gesture for mobile robots

PLOS ONE

Dear Dr. Reinhardt,

Thank you for submitting your manuscript to PLOS ONE. After careful consideration, we feel that it has merit but does not fully meet PLOS ONE’s publication criteria as it currently stands. Therefore, we invite you to submit a revised version of the manuscript that addresses the points raised during the review process.

Please address all the reviewer's concern and revise the manuscript again.

We look forward to receiving your revised manuscript.

Kind regards,

Quan Yuan, Ph.D.

Academic Editor

PLOS ONE

Reviewers' comments:

Reviewer's Responses to Questions

**Comments to the Author**

1. If the authors have adequately addressed your comments raised in a previous round of review and you feel that this manuscript is now acceptable for publication, you may indicate that here to bypass the “Comments to the Author” section, enter your conflict of interest statement in the “Confidential to Editor” section, and submit your "Accept" recommendation.

Reviewer #1: All comments have been addressed

Reviewer #2: All comments have been addressed

2. Is the manuscript technically sound, and do the data support the conclusions?

Reviewer #1: Yes

Reviewer #2: Yes

3. Has the statistical analysis been performed appropriately and rigorously? 

Reviewer #1: Yes

Reviewer #2: Yes

4. Have the authors made all data underlying the findings in their manuscript fully available?

Reviewer #1: Yes

Reviewer #2: Yes

5. Is the manuscript presented in an intelligible fashion and written in standard English?

Reviewer #1: Yes

Reviewer #2: Yes

6. Review Comments to the Author

Reviewer #1: My comments have been addressed. Given the paper in current shape, I have no further comments to provide.

Reviewer #2: 1. It is recommended to add and condense the innovation points in the introduction and list them clearly. What specific method is used to design a hesitant movement gesture for mobile robots？What's the difference and innovation？

2. Why is execution time considered a promising parameter? Please give a clear explanation.

7. PLOS authors have the option to publish the peer review history of their article (what does this mean?). If published, this will include your full peer review and any attached files.

Reviewer #1: No

Reviewer #2: No

---

## [Author Response · Author response to Decision Letter 1]

18 Feb 2021

The response is described in the file "Response to Reviewers.pdf".

---

## [Decision Letter · Decision Letter 2]

11 Mar 2021

Design of a hesitant movement gesture for mobile robots

PONE-D-20-27433R2

Dear Dr. Reinhardt,

We’re pleased to inform you that your manuscript has been judged scientifically suitable for publication and will be formally accepted for publication once it meets all outstanding technical requirements.

Kind regards,

Quan Yuan, Ph.D.

Academic Editor

PLOS ONE

Additional Editor Comments (optional):

Reviewers' comments:

Reviewer's Responses to Questions

**Comments to the Author**

1. If the authors have adequately addressed your comments raised in a previous round of review and you feel that this manuscript is now acceptable for publication, you may indicate that here to bypass the “Comments to the Author” section, enter your conflict of interest statement in the “Confidential to Editor” section, and submit your "Accept" recommendation.

Reviewer #2: All comments have been addressed

2. Is the manuscript technically sound, and do the data support the conclusions?

Reviewer #2: Yes

3. Has the statistical analysis been performed appropriately and rigorously? 

Reviewer #2: Yes

4. Have the authors made all data underlying the findings in their manuscript fully available?

Reviewer #2: Yes

5. Is the manuscript presented in an intelligible fashion and written in standard English?

Reviewer #2: Yes

6. Review Comments to the Author

Reviewer #2: (No Response)

7. PLOS authors have the option to publish the peer review history of their article (what does this mean?). If published, this will include your full peer review and any attached files.

Reviewer #2: No

---

## [Editor Report · Acceptance letter]

15 Mar 2021

PONE-D-20-27433R2 

Design of a hesitant movement gesture for mobile robots 

Dear Dr. Reinhardt:

I'm pleased to inform you that your manuscript has been deemed suitable for publication in PLOS ONE. Congratulations! Your manuscript is now with our production department. 

Kind regards, 

on behalf of

Dr. Quan Yuan 

Academic Editor

PLOS ONE